# Multi-Relational Structural Entropy

**Yuwei Cao**[1]    **Hao Peng** [*2]    **Angsheng Li**[2]    **Chenyu You**[3]    **Zhifeng Hao**[4]    **Philip S. Yu**[1]

[1]University of Illinois Chicago, Chicago, USA
[2]Beihang University, Beijing, China
[3]Yale University, New Haven, USA
[4]Shantou University, Shantou, China

## Abstract

Structural Entropy (SE) measures the structural information contained in a graph. Minimizing or maximizing SE helps to reveal or obscure the intrinsic structural patterns underlying graphs in an interpretable manner, finding applications in various tasks driven by networked data. However, SE ignores the heterogeneity inherent in the graph relations, which is ubiquitous in modern networks. In this work, we extend SE to consider heterogeneous relations and propose the first metric for multi-relational graph structural information, namely, multi-relational structural entropy (MrSE). To this end, we first cast SE through the novel lens of the stationary distribution from random surfing, which readily extends to multi-relational networks by considering the choices of both nodes and relation types simultaneously at each step. The resulting MrSE is then optimized by a new greedy algorithm to reveal the essential structures within a multi-relational network. Experimental results highlight that the proposed MrSE offers a more insightful interpretation of the structure of multi-relational graphs compared to SE. Additionally, it enhances the performance of two tasks that involve real-world multi-relational graphs, including node clustering and social event detection.

## 1 INTRODUCTION

In recent decades, graphs have become ubiquitous in our daily lives, with examples ranging from social networks and recommendation networks to publication networks, all effectively represented using graphs. Structural entropy (SE) [Li and Pan, 2016], which measures the amount of structural information contained in a graph, provides a useful tool

*Corresponding author

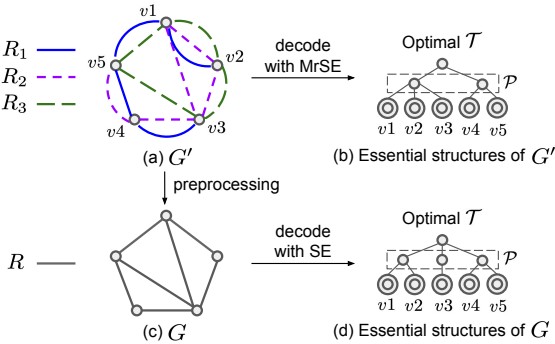

Figure 1: Decode the essential structures of a multi-relational graph with MrSE (ours) and SE. (a) is a multi-relational graph $G'$. (b) shows the essential structures of $G'$, decoded with MrSE. (c) is a single-relational graph $G$ reduced from $G'$. (d) shows the essential structures of $G$, decoded with SE.

for graph analysis. Specifically, unlike various graph measures [Raychaudhury et al., 1984, Braunstein et al., 2006, Dehmer, 2008, Bianconi, 2009] that are based on unstructured probability distributions, SE is interpretable [Liu et al., 2021]. Minimizing or maximizing SE helps to disclose or obfuscate the essential structures underlying the raw, noisy graphs. Such favorable properties of SE lead to its recent applications in tasks including graph pooling [Wu et al., 2022], community structure deception [Liu et al., 2019], graph contrastive learning [Wu et al., 2023], graph similarity measure [Liu et al., 2021], graph structure learning [Zou et al., 2023], network design [Liu et al., 2021], and social event detection [Cao et al., 2024].

However, SE assumes the existence of only a single type of relation between nodes, while in reality, graphs are multiplex [De Domenico et al., 2013] in nature, incorporating heterogeneous relation types. For example, there may be multiple edges between two papers in a publication network,

indicating shared authors, keywords, citations, accepted conferences, etc. As shown in Figures 1(a) and 1(c), analyzing multi-relational graphs with SE requires preprocessing them into single-relational graphs, which leads to information loss. This is due to the complementary nature of various relation types in revealing the graph's structure, with some relations being more informative than others. For instance, to determine if one paper is a follow-up study of another, examining the citations in addition to the keywords proves beneficial, while relying on accepted conferences may not provide as much insight. Therefore, it is essential to extend SE to consider multiple relation types.

In this work, we propose the first metric for multi-relational graph structural information, namely, multi-relational structural entropy (MrSE). Specifically, the original definition of SE measures the minimum number of bits required to determine the code of the node that is accessible with *one step of random walk* on a single-relational graph $G$ [Li and Pan, 2016], calculated from node degree statistics. Inspired by this, we propose to interpret SE with the stationary distribution vector obtained through *random surfing* [Page et al., 1999], i.e., taking an infinite long random walk, on $G$. Continuing with this idea, we then introduce the definition of the MrSE, incorporating random surfing on a multi-relational graph $G'$. During this multi-relational random surfing process, we simultaneously consider the choices of node and relation type at each step. The resulting stationary distribution vectors, one for nodes and one for relations, are used for MrSE calculation. We further illustrate how our proposed MrSE metric can be used to decode essential structures, such as communities, within $G'$. Through experiments on synthetic graphs, we demonstrate that our proposed MrSE outperforms SE in interpreting the structure of multi-relational graphs. Additionally, MrSE improves the performance of two real-world multi-relational graph tasks, namely node clustering and social event detection.

Our paper makes the following contributions:

- We introduce MrSE, the first metric designed to quantify the structural information within multi-relational graphs. Extending the favorable properties of SE, MrSE addresses heterogeneous relation types and serves as an improved tool for measuring and interpreting complex multi-relational graph structures.

- We demonstrate how our proposed MrSE metric can decipher structures within multi-relational graphs. Introducing an algorithm for 2-dimensional (2D) MrSE minimization, we enable the detection of communities within multi-relational graphs.

- Experiments on synthetic graphs with varying total numbers of relations, sizes, and sparsities demonstrate that our proposed MrSE, in comparison to SE, offers a more insightful interpretation of the structure of multi-relational graphs. Notably, a greater reduction

is observed when employing MrSE for graph entropy minimization, indicating a more effective decoding of structural information. Furthermore, experiments on real-world multi-relational graph data show that MrSE improves the performance of two tasks, namely node clustering and social event detection.

# 2 RELATED WORKS AND BACKGROUND

**Entropy-based Graph Metrics.** Measuring graph complexity is an important issue in graph analysis. To tackle this, various entropy-based graph measures [Raychaudhury et al., 1984, Braunstein et al., 2006, Dehmer, 2008, Bianconi, 2009, Li and Pan, 2016] have been proposed. Each of these measures represents a distinct form of Shannon entropy designed for different types of distributions extracted from the graphs. For example, the von Neumann graph entropy [Braunstein et al., 2006] is defined as the Shannon entropy of the Laplacian spectra. In contrast to previous metrics, SE quantifies the Shannon entropy of degree statistics, providing interpretability from an algebraic perspective. Meanwhile, all these metrics are designed for single-relational graphs. Hence, there is pressing need of a metric that can assess the complexity of multi-relational graphs.

**Structural Entropy (SE).** Let $G = (V, E)$ be a single-relational graph, where $V$ is a set of nodes and $E$ is a set of edges. Assuming that the structure of $G$ can be represented with an *encoding tree*, the formal definitions of the encoding tree and SE are as follows:

**Definition 2.1.** *[Li and Pan, 2016] An encoding tree $\mathcal{T}$ is a tree that encodes a hierarchical partition of $V$. $\mathcal{T}$ satisfies: 1) Each node $\alpha$ in $\mathcal{T}$ is associated with a node subset $T_\alpha \subseteq V$. In particular, the root node $\lambda$ of $\mathcal{T}$ is associated with $V$, and any leaf node $\gamma$ in $\mathcal{T}$ is associated with a single node in $V$. 2) The node subsets associated with the children of $\alpha$ form a partition of $T_\alpha$. 3) Denote the height of $\alpha$ as $h(\alpha)$. Let $h(\gamma) = 0$ and $h(\alpha^-) = h(\alpha) + 1$, where $\alpha^-$ is the parent node of $\alpha$. $h(\mathcal{T}) = \max_{\alpha \in \mathcal{T}}\{h(\alpha)\}$ is the height of $\mathcal{T}$.*

**Definition 2.2.** *[Li and Pan, 2016] Given a single-relational graph $G$ and an encoding tree $\mathcal{T}$, the structural entropy (SE) of $G$ relative to $\mathcal{T}$ is*

$$\mathcal{H}^{\mathcal{T}}(G) = - \sum_{\alpha \in \mathcal{T}, \alpha \neq \lambda} \frac{g_\alpha}{\text{vol}(T_\lambda)} \log \frac{\text{vol}(T_\alpha)}{\text{vol}(T_{\alpha^-})}, \quad (1)$$

*where $g_\alpha$ is the summation of the degrees of the cut edges of $T_\alpha$, i.e., edges in $E$ that have exactly one endpoint in $T_\alpha$. For a directed $G$, $g_\alpha$ is the summation of the in-degrees of the nodes in $T_\alpha$. $\text{vol}(\cdot)$ stands for the volume, i.e., the sum of the (in-)degrees, of the associated node subset. E.g., $\text{vol}(T_\alpha)$, $\text{vol}(T_{\alpha^-})$, and $\text{vol}(T_\lambda)$ refer to the volume of $T_\alpha$, $T_{\alpha^-}$, and $V$, respectively.*

Note that an encoding tree $\mathcal{T}$ is essentially a description of a graph's structure. Meanwhile, the SE values reveal how well $\mathcal{T}$ captures the structures of the graph. $\mathcal{H}^{(k)}(G)$, the $k$-dimensional SE of $G$, is defined as $\mathcal{H}^{\mathcal{T}}(G)$ that associated with a $\mathcal{T}$ that satisfies $h(\mathcal{T}) = k$. For $k = 1$, the 1-dimensional (1D) SE, $\mathcal{H}^{(1)}(G)$, is equivalent to the Shannon entropy of the degree heterogeneity. $\mathcal{H}^{(1)}(G)$ is associated with a unique $\mathcal{T}$ of height 1, measuring the intrinsic information within $G$ without making assumptions about higher-order structures, such as communities. For $k > 1$, minimizing or maximizing $\mathcal{H}^{(k)}(G)$ is equivalent to seeking a $\mathcal{T}$ of height $k$ that reveals or hides the $k$-dimensional structures within $G$. E.g., Figures 1(c) and 1(d) visualize how minimizing the 2-dimensional (2D) SE reveals the 2D structures, i.e., communities, within $G$. The resulting $\mathcal{T}$ is optimal, i.e., corresponds to the minimum 2D SE. $\mathcal{P} = \{\alpha | \alpha \in \mathcal{T}, h(\alpha) = 1\}$ forms a partition of $V$ that highlight the communities within $G$. We provide more examples of encoding trees and the 2D SE minimization process in Appendix B.

SE has found various applications [Wu et al., 2022, 2023, Liu et al., 2021, 2019, Li et al., 2016, 2018, Cao et al., 2024, Peng et al., 2024, Zou et al., 2024, Zeng et al., 2024, 2023a,b,c]. However, SE assumes the existence of only a single type of relation between nodes. This limitation calls for an improved SE measure that addresses the widespread heterogeneity of relation types.

# 3 MULTI-RELATIONAL STRUCTURAL ENTROPY (MRSE)

In this section, we first interpret SE from the perspective of random surfing (Section 3.1). Following this intuition, we then draw inspiration from random surfing on multi-relational networks and derive the multi-relational structural entropy (MrSE) measure (Section 3.2). Finally, we show how to decode the structures within a multi-relational graph by minimizing the proposed MrSE (Section 3.3). Appendix A summarizes the notations used in this work.

## 3.1 A RANDOM SURFING-BASED INTERPRETATION OF SE

As shown in Definition 2.2, the original definition of SE is based on degree statistics. Meanwhile, we observe that the degree heterogeneity of $G$ resembles the stationary probability vector resulting from *random surfing* [Page et al., 1999] on $G$. Leveraging this, we interpret Definition 2.2 through the lens of random surfing, as outlined below.

We denote the adjacency matrix of $G$ as $\mathbf{A} \in \mathbb{R}_+^{|V| \times |V|}$, with entry $\mathbf{A}_{j,i}$ equal the weight of the edge that starts from node $i \in V$ and ends at node $j \in V$. We assume that $G$ is strongly connected, i.e., $\mathbf{A}$ is irreducible. A surfer ran-

domly starts from node $i \in V$. At each step, the surfer randomly steps into a neighboring node in $\{j | \mathbf{A}_{j,i} \neq 0\}$. The surfing process can thus be seen as a Markov chain with transition probability matrix $\tilde{\mathbf{A}}$, where $\tilde{\mathbf{A}}$ is acquired from column-normalizing $\mathbf{A}$ such that $\forall i, \sum_{j=1}^{|V|} \tilde{\mathbf{A}}_{j,i} = 1$ holds. Let $\mathbf{x} \in \mathbb{R}_+^{|V|}$ be the stationary distribution, i.e., a probability distribution that indicates where the surfer is likely to be after an infinitely long walk. $\mathbf{x}$ satisfies $\mathbf{x} = \tilde{\mathbf{A}}\mathbf{x}$ and can be calculated using the Power Method [Journée et al., 2010]. With $\tilde{\mathbf{A}}$ and $\mathbf{x}$, Equation (1) can be rewritten as:

$$\mathcal{H}^{\mathcal{T}}(G) = - \sum_{\alpha \in \mathcal{T}, \alpha \neq \lambda} p_{\to \alpha} \log \frac{p_\alpha}{p_{\alpha^-}}, \qquad (2)$$

where $p_{\to \alpha} = \sum_{i \in V \setminus T_\alpha} \mathbf{x}_i \sum_{j \in T_\alpha} \tilde{\mathbf{A}}_{j,i}$, $p_\alpha = \sum_{i \in T_\alpha} \mathbf{x}_i$, and $p_{\alpha^-} = \sum_{i \in T_{\alpha^-}} \mathbf{x}_i$.

Particularly, the 1D SE of $G$ is rewritten as $\mathcal{H}^{(1)}(G) = -\sum_{i=1}^{|V|} \mathbf{x}_i \log \mathbf{x}_i$, which measures the intrinsic information contained in $G$.

**Proposition 3.1.** *Equation (1) and Equation (2) give the same definition of $\mathcal{H}^{\mathcal{T}}(G)$.*

*Proof.* For the first multiplicand on the RHS of Equation (1), we have

$$\begin{aligned} \frac{g_\alpha}{\text{vol}(T_\lambda)} &= \sum_{i \in V \setminus T_\alpha} \sum_{j \in T_\alpha} P(i, j) \\ &= \sum_{i \in V \setminus T_\alpha} P(i) \sum_{j \in T_\alpha} P(j|i) \\ &= \sum_{i \in V \setminus T_\alpha} \mathbf{x}_i \sum_{j \in T_\alpha} \tilde{\mathbf{A}}_{j,i} \\ &= p_{\to \alpha}. \end{aligned} \qquad (3)$$

Additionally, for the second multiplicand on the RHS of Equation (1), we have

$$\begin{aligned} \log \frac{\text{vol}(T_\alpha)}{\text{vol}(T_{\alpha^-})} &= \log \frac{\text{vol}(T_\alpha)}{\text{vol}(T_\lambda)} - \log \frac{\text{vol}(T_{\alpha^-})}{\text{vol}(T_\lambda)} \\ &= \log(\sum_{i \in T_\alpha} \mathbf{x}_i) - \log(\sum_{i \in T_{\alpha^-}} \mathbf{x}_i) \quad (4) \\ &= \log \frac{p_\alpha}{p_{\alpha^-}}, \end{aligned}$$

which concludes the proof. $\square$

The assumption of strong connectivity for $G$ may be violated in certain situations. In such cases, *stochasticity adjustment* [Page et al., 1999] is required to transform $G$ into a strongly connected graph. Specifically, we replace all zero columns in $\tilde{\mathbf{A}}$ with $1/|V|\mathbf{e}$, where $\mathbf{e}$ is a vector of ones. In addition, we make *primitivity adjustment* [Page et al., 1999] to decrease the number of iterations needed for the Power Method to converge. Specifically, we replace $\tilde{\mathbf{A}}$ with a new transition matrix $\mathbf{B} = c\tilde{\mathbf{A}} + (1 - c)\mathbf{E}$, where $(1 - c)$ is the probability

for the surfer to teleport to a random node and $\mathbf{E}$ is the teleportation matrix. We set $\mathbf{E}$ to $1/|V|\mathbf{e}\mathbf{e}^\top$ and $c$ to 0.85 following [Page et al., 1999]. We note, nonetheless, that the selection of $c$ requires balancing two demands: 1) $c$ is small enough so that the Power Method converges fast and 2) $c$ is reasonably large so that $G$ is not over-modulated and its intrinsic structural information is kept. We propose to explore the best strategy for choosing $c$ in the future.

## 3.2 FROM MULTI-RELATIONAL RANDOM SURFING TO MRSE

Following the intuitions in Section 3.1, we derive the first metric for multi-relational graph structural information, i.e., multi-relational structural entropy (MrSE), based on random surfing on multi-relational networks.

We denote a multi-relational network as $G' = (V, E', R)$, where $V$, $E'$, and $R$ stand for the node, edge, and relation sets of $G'$, respectively. The adjacency tensor of $G'$ is $\mathbf{A}' \in \mathbb{R}_+^{|V|\times|V|\times|R|}$, with entry $\mathbf{A}'_{i,j,r}$ equals the weight of the edge that starts from $j \in V$, ends at $i \in V$, and associates with relation $r \in R$. At each step of the multi-relational surfing, the surfer jointly considers which neighboring *node* to visit and which *relation* to use. We provide examples of $G'$, $\mathbf{A}'$, and multi-relational random surfing in Appendix C. Inspired by [Ng et al., 2011], we use two transition matrices, denoted as $\mathcal{V}$ and $\mathcal{R}$, to model the choices of the neighboring node and the relation, respectively. We assume that $\mathbf{A}'$ is irreducible [Ng et al., 2011], i.e., for any fixed $r$, a slice of $\mathbf{A}'$, $(\mathbf{A}'_{i,j,r}) \in \mathbb{R}_+^{|V|\times|V|}$ is irreducible. $\mathcal{V}$ and $\mathcal{R}$ are constructed as $\mathcal{V}_{i,j,r} = \mathbf{A}'_{i,j,r}/\sum_{i=1}^{|V|}\mathbf{A}'_{i,j,r}$ and $\mathcal{R}_{i,j,r} = \mathbf{A}'_{i,j,r}/\sum_{r=1}^{|R|}\mathbf{A}'_{i,j,r}$, respectively. Let $\mathbf{x}' \in \mathbb{R}_+^{|V|}$ and $\mathbf{y} \in \mathbb{R}_+^{|R|}$ be two probability distributions that tell us which node the surfer is likely to visit and which relation the surfer is likely to use at each step, respectively. After an infinitely long walk on $G'$, $\mathbf{x}'$ and $\mathbf{y}$ would converge to two stationary distributions that satisfy $\sum_{r=1}^{|R|}\sum_{j=1}^{|V|}\mathcal{V}_{i,j,r}\mathbf{x}'_j\mathbf{y}_r = \mathbf{x}'_i$ and $\sum_{i=1}^{|V|}\sum_{j=1}^{|V|}\mathcal{R}_{i,j,r}\mathbf{x}'_j\mathbf{x}'_i = \mathbf{y}_r$, respectively. $\mathbf{x}'$ and $\mathbf{y}$ can be calculated using the MultiRank algorithm [Ng et al., 2011]. With $\mathcal{V}$, $\mathbf{x}'$, and $\mathbf{y}$, we introduce the definition of MrSE as follows.

**Definition 3.2.** *Given a multi-relational graph $G'$, and an encoding tree $\mathcal{T}$. Assume we have the node and relation stationary distributions $\mathbf{x}'$ and $\mathbf{y}$ acquired from multi-relational random surfing on $G'$ following node and relation transition matrices $\mathcal{V}$ and $\mathcal{R}$. The multi-relational structural entropy (MrSE) of $G'$ relative to $\mathcal{T}$ is*

$$\mathcal{H}^{\mathcal{T}}(G') = -\sum_{\alpha \in \mathcal{T}, \alpha \neq \lambda} p'_{\to\alpha} \log \frac{p'_\alpha}{p'_{\alpha^-}}, \tag{5}$$

where $p'_{\to\alpha} = \sum_{i\in V\backslash T_\alpha}\mathbf{x}'_i\sum_{j\in T_\alpha}\sum_{r\in R}\mathcal{V}_{j,i,r}\mathbf{y}_r$, $p'_\alpha = \sum_{i\in T_\alpha}\mathbf{x}'_i$, and $p'_{\alpha^-} = \sum_{i\in T_{\alpha^-}}\mathbf{x}'_i$.

Particularly, the 1D MrSE of $G'$, $\mathcal{H}^{(1)}(G') = -\sum_{i=1}^{|V|}\mathbf{x}'_i\log\mathbf{x}'_i$, measures the intrinsic information contained in $G'$.

**Proposition 3.3.** *The probabilistic interpretations of SE and MrSE are identical.*

*Proof.* Both $p'_{\to\alpha}$ in Equation (5) and $p_{\to\alpha}$ in Equation (2) essentially mean the probability of entering community $T_\alpha$. We have

$$\begin{aligned}
p'_{\to\alpha} &= \sum_{i\in V\backslash T_\alpha}\sum_{j\in T_\alpha}P(i,j)\\
&= \sum_{i\in V\backslash T_\alpha}P(i)\sum_{j\in T_\alpha}P(j|i)\\
&= \sum_{i\in V\backslash T_\alpha}P(i)\sum_{j\in T_\alpha}\sum_{r\in R}P(j|i,r)P(r)\\
&= \sum_{i\in V\backslash T_\alpha}\mathbf{x}'_i\sum_{j\in T_\alpha}\sum_{r\in R}\mathcal{V}_{j,i,r}\mathbf{y}_r,
\end{aligned} \tag{6}$$

so it aligns with the probabilistic interpretation of $p_{\to\alpha}$ as shown in Equation (3).

Additionally, $p'_\alpha$ and $p'_{\alpha^-}$ in Equation (5) stand for the probabilities of the surfer being in communities $T_\alpha$ and $T_{\alpha^-}$, respectively. Since $T_\alpha \subset T_{\alpha^-}$, the surfer has to be already in community $T_{\alpha^-}$ before they can enter $T_\alpha$. Therefore, $\log\frac{p'_\alpha}{p'_{\alpha^-}} = \log p'_\alpha - \log p'_{\alpha^-}$ is the amount of *new* information, measured in bits, contained in entering $T_\alpha$. Similarly, $\log\frac{p_\alpha}{p_{\alpha^-}}$ in Equation (2) also stands for the new information contained in entering $T_\alpha$.

Consequently, SE measures the amount of information contained in one step of random walk on a single-relational $G$, while MrSE is the multi-relational counterpart of SE. MrSE and SE share the same probabilistic interpretation. $\qquad\square$

In the case that $\mathbf{A}'$ is reducible, we need to adjust $G'$ to ensure that $\mathbf{x}'$ and $\mathbf{y}$ converge. Specifically, we make stochasticity adjustments to $\mathcal{V}$ and $\mathcal{R}$ such that $\forall(i,r), \sum_{j=1}^{|V|}\mathcal{V}_{j,i,r} = 1$ and $\forall(j,i), \sum_{r=1}^{|R|}\mathcal{R}_{j,i,r} = 1$. Additionally, for faster convergence, we make primitivity adjustment by replacing $\mathcal{V}$ with $c\mathcal{V}+(1-c)\mathbf{E}'$, where $\mathbf{E}'$ is the teleportation matrix. We set $c$ to 0.85 and $\mathbf{E}' = 1/|V|\mathbf{1}$, where $\mathbf{1}$ is a $|V|\times|V|\times|R|$ all-ones matrix. These choices follow the same intuition as Section 3.1. Specifically, the $\mathbf{E}'$ value specifies that for any relation $r \in R$, the surfer has equal chances to teleport to any of the objects.

## 3.3 DECODING MULTI-RELATIONAL GRAPH STRUCTURE VIA MRSE MINIMIZATION

Uncovering the essential structures within the raw and noisy graphs is crucial. 2D SE minimization [Li and Pan, 2016] provides an effective unsupervised tool for decoding communities from single-relational graphs and has found applications in various tasks [Wu et al., 2022, 2023, Cao et al., 2024]. In this section, we propose to reveal the essential structures within multi-relational graphs by minimizing the proposed MrSE metric [1].

Firstly, following [Li and Pan, 2016], a MERGE operator is defined as follows.

**Definition 3.4.** *Given an encoding tree $\mathcal{T}$ and its two non-root nodes, $\alpha_{o_1}$ and $\alpha_{o_2}$, $MERGE(\alpha_{o_1}, \alpha_{o_2})$ removes $\alpha_{o_1}$ and $\alpha_{o_2}$ from $\mathcal{T}$ and adds a new node $\alpha_n$ to $\mathcal{T}$. $\alpha_n$ satisfies: 1) the children nodes of $\alpha_n$ in $\mathcal{T}$ is a combination of the children of $\alpha_{o_1}$ and $\alpha_{o_2}$; 2) $\alpha_n^- = \lambda$.*

The merge operation changes $\mathcal{T}$ and, therefore, would cause a change in the associated MrSE value. Based on Definition 3.2, the change follows:

$$\Delta \mathrm{MrSE}_{\alpha_{o_1}, \alpha_{o_2}} = \mathrm{MrSE}_{new} - \mathrm{MrSE}_{old}$$

$$= -p'_{\to \alpha_n} \log p'_{\alpha_n} - p'_{\alpha_{o_1}} \log \frac{p'_{\alpha_{o_1}}}{p'_{\alpha_n}} - p'_{\alpha_{o_2}} \log \frac{p'_{\alpha_{o_2}}}{p'_{\alpha_n}} \quad (7)$$

$$+ p'_{\to \alpha_{o_1}} \log p'_{\alpha_{o_1}} + p'_{\to \alpha_{o_2}} \log p'_{\alpha_{o_2}}.$$

The derivation of Equation (7) can be found in Appendix E.

We propose a 2D MrSE minimization algorithm, as shown in Algorithm 1. Initially, the encoding tree $\mathcal{T}$ assumes no communities, and each node $v \in V$ is assigned to its own cluster (line 2). At this point, the 2D MrSE associated with $\mathcal{T}$ equals the 1D MrSE and represents the intrinsic structural information within $G'$. Subsequently, the minimum 2D MrSE can be achieved by greedily and repeatedly merging the two tree nodes in $\mathcal{T}$ that would result in the largest $|\Delta \mathrm{MrSE}|$ until no further merge can lead to a $\Delta \mathrm{MrSE} < 0$ (lines 3 - 17). The algorithm outputs an optimized $\mathcal{T}$, associated with the minimum 2D MrSE. At this time, $\mathcal{T}$ encodes reliable structures within $G'$ while eliminating the noisy ones. Specifically, $\mathcal{P}$, the set of the tree nodes of height one, forms a partition of $V$ that reveals the communities in $G'$. Figures 1(a) and 1(b) visualize the 2D MrSE minimization process. We also provide a more detailed visualization in Appendix C. In addition, we note that higher-dimensional (such as 3D) MrSE minimization can be realized by repeatedly applying our 2D MrSE minimization algorithm and consolidating the identified communities into nodes.

---

[1] Meanwhile, we note that some tasks instead require concealing the essential structures within graphs, i.e., maximizing MrSE. One such example is community structure deception [Liu et al., 2019]. We defer the investigation of MrSE maximization algorithms to the future as they typically require task-specific design.

**Time Complexity.** The multi-relational random surfing in line 1 costs $O(|E'|)$ [Ng et al., 2011]. The construction of initial $\mathcal{T}$ in line 2 takes $O(|V|)$. The while loop in lines 3 - 17 takes $O(|V||E'|)$. The overall time complexity of Algorithm 1 is thus $O(|V||E'|)$.

**Hierarchical 2D MrSE Minimization.** Additionally, we note that hierarchical graph partitioning can be integrated to expedite the algorithm. Inspired by the hierarchical 2D SE minimization [Cao et al., 2024], we introduce a hierarchical 2D MrSE minimization algorithm (Appendix D) that takes $O(n^3)$. Hyperparameter $n$ is the size of the subgraph under consideration at each iteration and can be set to $o(|V|)$.

---

**Algorithm 1:** 2D MrSE minimization

**Input:** Multi-relational graph $G' = (V, E', R)$
**Output:** An optimal encoding tree $\mathcal{T}$ of height 2

1  Acquire the node and relation stationary distributions $\mathbf{x}'$ and $\mathbf{y}$ from multi-relational random surfing on $G'$
2  Initialize $\mathcal{T}$, s.t. for each node $v \in V$, add two nodes $\alpha$ and $\alpha^-$ to $\mathcal{T}$. $\alpha$ is a leaf node of $\mathcal{T}$ and $T_\alpha = \{v\}$. $\alpha^-$ is the parent of $\alpha$ and $h(\alpha^-) = 1$
3  **while** *True* **do**
4     $\mathcal{P} \leftarrow (\alpha | \alpha \in \mathcal{T}, h(\alpha) = 1)$
5     $\Delta \mathrm{MrSE} \leftarrow \infty$
6     **for** $\alpha_i \in \mathcal{P}$ **do**
7       **for** $\alpha_j \in \mathcal{P}, j > i$ **do**
8         **if** *there are edges between $T_{\alpha_i}$ and $T_{\alpha_j}$* **then**
9           $\Delta \mathrm{MrSE}_{ij} \leftarrow$ Eq. (7), w/o actually merging $\alpha_i$ and $\alpha_j$
10          **if** $\Delta MrSE_{ij} < \Delta \mathrm{MrSE}$ **then**
11            $\Delta \mathrm{MrSE} = \Delta \mathrm{MrSE}_{ij}$
12            $\alpha_{o_1} = \alpha_i$
13            $\alpha_{o_2} = \alpha_j$
14    **if** $\Delta MrSE < 0$ **then**
15      $\mathrm{MERGE}(\alpha_{o_1}, \alpha_{o_2})$
16    **else**
17      Break
18 **return** $\mathcal{T}$

---

## 4 EXPERIMENTS

We show that the proposed random surfing-based SE (denoted as RSSE for simplicity) approximates the original SE. Moreover, we show that as compared to SE, MrSE provides a better metric for multi-relational graph structures. We experiment on synthetic graphs (Section 4.1) as well as tasks that involve real-world multi-relational graphs, namely node clustering (Section 4.2) and social event detection (Section

4.3). Our code is publicly available [2].

## 4.1 SIMULATION EXPERIMENTS

We conduct a study on single- and multi-relational synthetic graphs generated using the Barabasi-Albert (BA) model [Albert and Barabási, 2002]. The BA model incorporates two important general concepts that exist widely in real-world networks: growth, i.e., the network increases over time, and preferential attachment, i.e., the more connected a node is, the more likely it is to receive new links. We proceed to describe our graph generation process.

**Synthetic data.** To generate single-relational graphs, we adopt the BA graph generator from PyG [3]. To regulate graph sparsity, edges are randomly dropped out to align with the desired sparsity level. To generate multi-relational graphs, we create multiple BA graphs of identical sizes and then concatenate them along the relation axis. Note that the BA graphs associated with each relation are generated independently, assuming no correlations between the relations.

**Experiment setup.** The calculation of 1D SE, RSSE, and MrSE follow Definition 2.2, Equation (2), and Definition 3.2, respectively. The calculation of the minimum 2D SE and RSSE follow the 2D SE minimization algorithm in [Li and Pan, 2016], while the calculation of the minimum MrSE follows the proposed 2D MrSE minimization algorithm (Algorithm 1). Note that SE, RSSE, and the 2D SE minimization algorithm are metrics and algorithms designed for single-relational graphs. To apply them on a multi-relational graph $G'$, we preprocess $G'$ by ignoring the heterogeneity in its relations and mapping $G'$ into a single-relational graph $G$, as visualized by Figures 1 (a) and 1 (c).

**Compare random surfing-based SE to original SE.** Figure 2 presents the 1D and the *minimum* 2D (denoted as 2D in the legend for simplicity) SE and RSSE of single-relational graphs with varying sizes and sparsities. In Figure 2 (a), the 1D SE and RSSE values increase with the graph size. This means larger graphs contain more structural information. In addition, the minimum 2D SE value also increases with the graph size, indicating that larger graphs contain more noise (this noise refers to the structural information that the optimal encoding tree, derived from the 2D SE minimization process, struggles to interpret). Meanwhile, the 1D RSSE values closely match the 1D SE values, and the minimum 2D RSSE values closely align with the minimum 2D SE values. This alignment suggests that our proposed random surfing-based method is a reliable approximation of the original SE. Likewise, Figure 2 (b) suggests that denser graphs encompass greater structural information and are more noisy. In addition, our proposed random surfing-based method effectively approximates the original SE, except for

[2]https://github.com/YuweiCao-UIC/MrSE.git
[3]https://pyg.org/

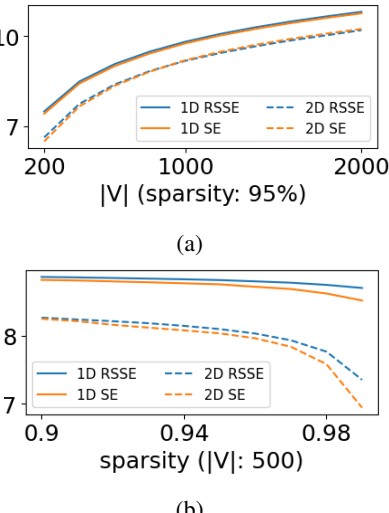

Figure 2: The 1D and 2D SE and RSSE of single-relational graphs with varying sizes (a) and sparsities (b).

very sparse (sparsity $> 98\%$) graphs. When the graph is sparse, both 1D and minimum 2D RSSE are higher than that of the original SE. This is caused by the imagined edges and the teleportation matrix introduced during the stochasticity and primitivity adjustments (detailed in Section 3.2).

**Decode multi-relational graph structural information.** We compare the effectiveness of MrSE, RSSE, and SE in decoding the structural information of multi-relational graphs. For each multi-relational graph $G'$, we measure $\Delta\text{MrSE} = (1\text{D MrSE} - \text{minimum 2D MrSE})/1\text{D MrSE}$, which represents the fraction of the structural information within $G'$ that successfully decoded by minimizing 2D MrSE. The larger $\Delta\text{MrSE}$ is, the more effective MrSE is at deciphering the structure of $G'$. We measure $\Delta\text{SE}$ and $\Delta\text{RSSE}$ in similar manners, except that the heterogeneity in the relations of $G'$ is ignored, and 2D SE minimization [Li and Pan, 2016] instead of our proposed 2D MrSE minimization algorithm is applied. Figures 3(a), 3(b), and 3(a) present the $\Delta\text{SE}$, $\Delta\text{RSSE}$, and $\Delta\text{MrSE}$ of multi-relational graphs with varying sizes, total number of relations, and sparsities, respectively. We can tell that as the graph size and total number of relations increase, $\Delta\text{SE}$, $\Delta\text{RSSE}$, and $\Delta\text{MrSE}$ show a declining pattern, while they exhibit an ascending trend with sparsity. This suggests that graphs that are larger, denser, and contain more complex relations are more difficult to decipher. Moreover, it is evident that $\Delta\text{MrSE}$ consistently surpasses $\Delta\text{SE}$ and $\Delta\text{RSSE}$, despite the changes in graph size, total number of relations, and sparsity. This suggests that our proposed MrSE, compared to SE and RSSE, offers a more effective tool for measuring and decoding the structural information in multi-relational graphs.

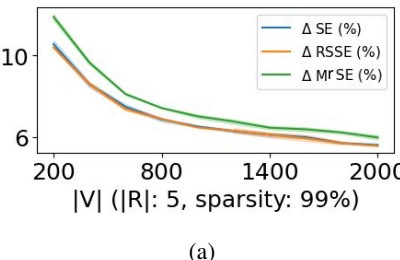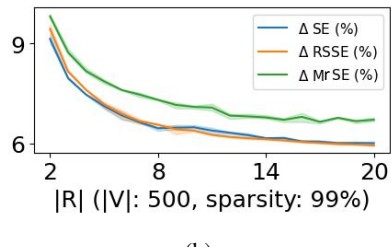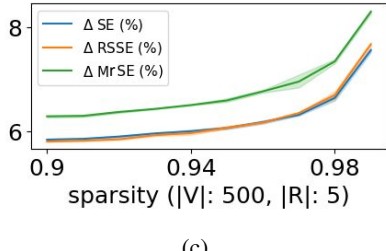

|   |   |   |
|:-:|:-:|:-:|
| (a) | (b) | (c) |

Figure 3: The $\Delta$SE, $\Delta$RSSE, and $\Delta$MrSE of multi-relational graphs with varying sizes (a), the total number of relations (b), and sparsities (c).

Table 1: Statistics of the node clustering datasets. For IMDB, M, A, and D denote movie, actor, and director; for DBLP, A, P, T, and C denote author, paper, term, and conference; for ACM, P, C, A, S, and T denote paper, cite, author, subject, and term.

| Dataset | $|V|$ | $R$ | $|E'|$ | Sparsity (%) | $|Y|$ |
|:---|:---:|:---:|:---:|:---:|:---:|
| ACM | 3,025 | P-C-P | 5,335 | 99.88 | 3 |
|  |  | P-A-P | 13,374 | 99.71 |  |
|  |  | P-S-P | 1,107,032 | 75.80 |  |
|  |  | P-T-P | 4,573,785 | <0.01 |  |
| DBLP | 4,057 | A-P-A | 3,528 | 99.96 | 4 |
|  |  | A-P-T-P-A | 3,519,757 | 57.22 |  |
|  |  | A-P-C-P-A | 2,498,219 | 69.64 |  |
| IMDB | 4,278 | M-A-M | 40,540 | 99.56 | 3 |
|  |  | M-D-M | 6,584 | 99.93 |  |

## 4.2 MULTI-RELATIONAL NODE CLUSTERING

Unsupervised node clustering is an essential task in graph analysis. In this section, we evaluate our proposed MrSE on multi-relational graph node clustering.

**Datasets.** Following previous multi-relational graph embedding studies [Park et al., 2020], we evaluate on IMDB [Fu et al., 2020], DBLP [Fu et al., 2020], and ACM [Lv et al., 2021]. IMDB is a movie dataset. The movies are divided into three classes (action, comedy, drama) according to their genre. Movie features correspond to the bag-of-words representations of plots. DBLP is a publication dataset containing authors that are labeled according to their research areas (database, data mining, machine learning, information retrieval). Author features are the bag-of-words representations of keywords. ACM is a publication dataset containing papers divided into three classes (database, wireless communication, and data mining). Paper features correspond to the bag-of-words representations of keywords. Following [Park et al., 2020], the multi-relational edges are inferred via intermediate nodes (e.g., for IMDB, the 'M-A-M' edges are inferred via actor nodes, and the 'M-D-M' edges are inferred via director nodes). Table 2 shows the data statistics.

**Baselines.** We compare the proposed **MrSE** to **SE** [Li and Pan, 2016] and **RSSE**. We further compare to a deep learning-based spectral clustering method, i.e., **SpectralNet**

[Shaham et al., 2018]. We also consider random walk-based methods, including **node2vec** [Grover and Leskovec, 2016], which learns node embeddings with random walks and skip-gram, and **metapath2vec** [Dong et al., 2017], which performs metapath-based random walk. In addition, we compared to GNN-based methods, including **DGI** [Veličković et al., 2019], which maximizes global-local mutual information; **DMGI** [Park et al., 2020], which is the multi-relational counterpart of DGI; **DMoN** [Tsitsulin et al., 2023], which maximizes graph modularity [Newman, 2006]. We additionally perform **k-means** clustering using node features to gauge their informativeness. All methods are unsupervised. Among them, metapath2vec, DMGI, and MrSE explore the heterogeneity of relations. Note that the GNN-based methods, i.e., DGI, DMGI, and DMoN, leverage node features in addition to graph structure, which gives them an extra edge over methods that rely solely on graph structure.

**Metrics.** Following [Park et al., 2020, Tsitsulin et al., 2023], we report normalized mutual information (NMI). We further report adjusted rand index (ARI), and unsupervised clustering accuracy (ACC, in Appendix G), which are commonly adopted clustering metrics.

**Experiment setup.** To evaluate single-relational methods including SpecturalNet, DMoN, DGI, SE, and RSSE, we preprocess the multi-relational datasets by mapping them into single-relational ones. Additionally, following [Park et al., 2020], we explore leveraging heterogeneous relations with the single-relational embedding methods, i.e., SpecturalNet, DMoN, and DGI. Specifically, we obtain the final node embedding by averaging the node embeddings obtained from single-relational graphs that correspond to each relation. For node2vec and metapath2vec, we extend the graphs to contain the intermediate nodes (e.g., the actor and director nodes for IMDB). For k-means, we set the number of clusters to the ground truth, i.e., 3, 4, 3 for ACM, DBLP, and IMDB, respectively. Similarly, for the representation-learning models including node2vec, metapath2vec, DGI, and DMGI, we perform k-means clustering after learning representations, setting the number of clusters to the ground truth to obtain final community predictions. We implement SE, RSSE, and MrSE using Python. For SpectralNet, we

use the source code provided by the authors [4]. For the rest models, we leverage the implementations from the PyG package. We repeat all experiments 5 times and average results across runs. The method-specific hyperparameters are decided according to the original papers and are provided in Appendix F.

**Node clustering results.** Table 2 presents the node clustering results. MrSE outperforms all baselines in both NMI and ARI on the ACM and DBLP datasets. On the IMDB dataset, MrSE achieves the highest NMI but falls short in terms of ARI. These results highlight the strong capability of MrSE in identifying communities within multi-relational graphs. This is particularly noteworthy given that certain methods, i.e., the GNN-based ones, utilize graph structure and node features, whereas MrSE relies exclusively on graph structure.

For the ACM dataset, which contains extremely dense relations (e.g., 'P-T-P'), single-relational methods show near-zero results or fail to run. Addressing the heterogeneity of relations, either by averaging per-relation embeddings, introducing metapaths, or applying distinct weights to relations, results in a substantial performance boost. E.g., DGI with $G', X$ input outperforms DGI with $G, X$ input, metapath2vec outperforms node2vec, and MrSE outperforms SE and RSSE. For the DBLP and IMDB datasets, the strategy of averaging per-relation embeddings proves to be less effective in handling heterogeneous relations, as evidenced by the lower performance of DGI with $G', X$ input compared to DGI with $G, X$ input. Somewhat surprisingly, DMGI performs worse than DGI with $G', X$ input on all three datasets, suggesting that weighting per-relation embeddings with attention is less effective than simply averaging them.

In addition, we observe that SE and RSSE resemble each other across datasets and metrics, showing that RSSE effectively approximates SE. Moreover, MrSE consistently outperforms SE and RSSE by large margins, indicating that MrSE offers a better metric for interpreting the structural information within multi-relational graphs.

## 4.3 SOCIAL EVENT DETECTION

We note that the proposed MrSE, serving as the multi-relational counterpart to SE, can enhance the extensive applications of SE by tackling heterogeneous relations. One such application is social event detection, which is commonly formalized as extracting clusters of co-related messages from streams of social media messages. [Cao et al., 2024] achieves SOTA social event detection performance using 2D SE minimization. However, it overlooks the heterogeneity of message correlations. In this section, we explore social event detection using the proposed 2D MrSE minimization and observe the performance changes resulting from the

---

[4]https://github.com/shaham-lab/SpectralNet

Table 2: Node clustering results (%). '/' indicates that SpectralNet fails to run on ACM.

| Method | Input | ACM | | DBLP | | IMDB | |
|---|---|---|---|---|---|---|---|
| | | NMI | ARI | NMI | ARI | NMI | ARI |
| k-means | $X$ | 25.80 | 16.32 | 20.65 | 7.37 | 3.59 | 0.00 |
| DMoN | $G, X$ | 0.00 | 0.00 | 38.23 | 6.50 | 4.53 | 0.61 |
| DGI | $G, X$ | 0.32 | 0.01 | 33.38 | 12.85 | 7.70 | **9.07** |
| SpectralNet | $G$ | / | / | 39.09 | 8.44 | 3.83 | 0.69 |
| node2vec | $G$ | 0.09 | 0.03 | 27.04 | 15.90 | 2.75 | 3.01 |
| SE | $G$ | 3.16 | 3.16 | 39.10 | _41.84_ | **8.82** | 0.17 |
| RSSE (ours) | $G$ | 3.56 | 3.66 | 39.12 | 41.06 | **8.82** | 0.18 |
| DMoN | $G', X$ | 21.10 | 8.82 | 15.99 | 6.84 | 1.27 | 0.47 |
| DGI | $G', X$ | 47.03 | _43.66_ | 34.29 | 29.79 | 0.44 | 0.00 |
| DMGI | $G', X$ | 25.53 | 20.50 | 1.43 | 1.21 | 0.63 | 0.45 |
| SpectralNet | $G'$ | / | / | 36.58 | 30.88 | 1.16 | 0.15 |
| metapath2vec | $G'$ | _47.51_ | 42.90 | _45.56_ | 36.76 | 5.34 | _5.10_ |
| MrSE (ours) | $G'$ | **48.36** | **55.80** | **49.26** | **55.78** | **13.68** | 0.19 |

Table 3: Social event detection results (%), averaged over all blocks.

| Dataset | Metric | HISEvent | RSSE (ours) | MrSE (ours) |
|---|---|---|---|---|
| Event2012 | NMI | 82.94 | _83.01_ | **84.17** |
| | ARI | _63.15_ | 63.10 | **64.17** |
| Event2018 | NMI | _76.08_ | 75.82 | **76.64** |
| | ARI | _60.25_ | 59.39 | **60.91** |

introduction of heterogeneous message correlations.

**Datasets.** We experiment on two large, public Twitter datasets, i.e., Event2012 [McMinn et al., 2013] and Event2018 [Mazoyer et al., 2020]. Within Event2012, there are 68,841 English tweets associated with 503 events, spanning a four-week period. Event2018 consists of 64,516 French tweets discussing 257 events and occurring over a 23-day period. We adopt the data splits of [Cao et al., 2024] to evaluate under the open-set settings, which assumes the events happen over time and splits the datasets into day-wise message blocks. Data statistics are in Appendix H.

**Baselines.** We compare to **HISEvent** [Cao et al., 2024], which is the current SOTA of social event detection. It begins by constructing *message graphs*, where nodes represent messages and correlated messages are connected (these correlations may arise from shared senders, mentioned users, hashtags, named entities, or similar natural language semantics. Such heterogeneity is ignored). Subsequently, it partitions the message graphs using 2D SE minimization to extract social events, which are represented by clusters of messages. Note that we omit the direct comparison with various social event detectors that HISEvent has outperformed, including ones that leverage GNN [Cao et al., 2021, Ren et al., 2022, Peng et al., 2022, Ren et al., 2023], betweenness centrality [Liu et al., 2020], TF-IDF [Bafna et al., 2016], LDA [Blei et al., 2003], etc.

**Metrics.** Following previous social event detection studies [Cao et al., 2024], we report NMI and ARI.

**Experiment setup.** For HISEvent, we use the source code

provided by the authors [5]. To evaluate RSSE, we simply replace the SE in HISEvent with RSSE. To evaluate MrSE, we make two changes: first, we explore the heterogeneity of message correlations and construct *multi-relational message graphs* (detailed in Appendix I); second, we replace the 2D SE minimization in HISEvent with the proposed 2D MrSE minimization. For all three methods, we adopt the same hyperparameters as specified in the HISEvent paper.

**Social event detection results.** Table 3 presents the social event detection results. MrSE demonstrates superior performance compared to HISEvent across datasets and metrics. This suggests that by delving into heterogeneous message correlations, the proposed MrSE enhances the efficacy of social event detection in comparison to HISEvent, which relies on the original SE and overlooks the heterogeneity in message correlations. Furthermore, RSSE performs comparably to HISEvent, which utilizes SE. This suggests that RSSE and SE can be used interchangeably.

## 5 CONCLUSION

In this study, we propose MrSE, the first metric of multi-relational graph structural information. We begin by reexamining the original definition of SE from the viewpoint of random surfing. Subsequently, the definition of MrSE is derived from random surfing on multi-relational graphs. Additionally, we introduce a 2D MrSE minimization algorithm designed to unveil communities within these complex graphs. Results from experiments on both synthetic and real-world graphs, including movie, publication, and social message networks, demonstrate that the proposed MrSE is a powerful metric for assessing and unraveling the structural information within multi-relational graphs. MrSE exhibits strong performance in two tasks, namely multi-relational node clustering and social event detection.

## ACKNOWLEDGMENTS

We thank Prof. Xinhua Zhang for his helpful discussions and suggestions. This work is supported by the National Key R&D Program of China through grant 2022YFB3104700, NSFC through grants 62322202 and 61932002, Beijing Natural Science Foundation through grant 4222030, Guangdong Basic and Applied Basic Research Foundation through grant 2023B1515120020, S&T Program of Hebei through grants 20310101D and 21340301D, and Shijiazhuang Science and Technology Plan Project through grant 231130459A. Philip S. Yu is supported in part by NSF under grant III-2106758.

---

[5]https://github.com/SELGroup/HISEvent

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

| Notation | Description |
|---|---|
| $G = (V, E)$ | Single-relational graph with node set $V$ and edge set $E$ |
| $\mathbf{A}$ | $\mathbf{A} \in \mathbb{R}_+^{|V| \times |V|}$, the adjacency matrix of $G$ |
| $\tilde{\mathbf{A}}$ | Transition probability matrix of random surfing |
| $\mathbf{B}$ | $\tilde{\mathbf{A}}$ after primitivity adjustment |
| $\mathbf{E}$ | Teleportation matrix |
| $\mathbf{x}$ | The stationary distribution of random surfing |
| $G' = (V, E', R)$ | Multi-relational graph with node set $V$, heterogeneous edge set $E'$, and relation set $R$ |
| $\mathbf{A}'$ | $\mathbf{A}' \in \mathbb{R}_+^{|V| \times |V| \times |R|}$, the adjacency matrix of $G'$ |
| $\mathcal{V}; \mathcal{R}$ | Node and relation transition probability matrices of multi-relational random surfing |
| $\mathbf{E}$ | Multi-relational teleportation matrix |
| $\mathbf{x}'; \mathbf{y}$ | Node and relation stationary distributions of multi-relational random surfing |
| $\mathcal{T}$ | Encoding tree |
| $\alpha, \lambda, \gamma \in \mathcal{T}$ | Node, root node, leaf node in $\mathcal{T}$ |
| $\alpha^-$ | The parent of $\alpha$ |
| $T_\alpha, T_\lambda, T_\gamma \in V$ | Node sets $\subseteq V$ that associate with $\alpha, \lambda, \gamma$ |
| $h(\alpha)$; | Height of $\alpha$ |
| $h(\mathcal{T})$; | Height of $\mathcal{T}$ |
| $g_\alpha$ | Summation of the degrees of the cut edges of $T_\alpha$ |
| $\mathrm{vol}(T_\alpha); \mathrm{vol}(T_\lambda)$ | Volume of $T_\alpha$; Volume of $T_\lambda$, i.e., $V$ |
| $\mathcal{P}$ | A partition of $V$ |
| $\mathcal{H}^{\mathcal{T}}(G)$ | The structural entropy (SE) of $G$ relative to $\mathcal{T}$ |
| $\mathcal{H}^{(k)}(G)$ | The $k$-dimensional SE of $G$ |
| $\mathcal{H}^{\mathcal{T}}(G')$ | The multi-relational structural entropy (MrSE) of $G'$ relative to $\mathcal{T}$ |
| $\mathcal{H}^{(k)}(G')$ | The $k$-dimensional MrSE of $G'$ |
| $p_{\to\alpha}$ | The probability of entering community $T_\alpha$ during random surfing |
| $p_\alpha$ | The probability of being in community $T_\alpha$ during random surfing |
| $p'_{\to\alpha}$ | The probability of entering community $T_\alpha$ during multi-relational random surfing |
| $p'_\alpha$ | The probability of being in community $T_\alpha$ during multi-relational random surfing |

Table 4: Glossary of Notations.

# A NOTATIONS

Table 4 summarizes the main notations used in this paper.

# B EXAMPLES OF 2D SE MINIMIZATION

We provide examples of single-relational graph, encoding trees, and the 2D SE minimization process in Figure 4.

Note that an encoding tree is essentially a description of a graph's structure. For a graph $G$, the encoding tree of height 1 is unique, containing one root node and $|V|$ leaf nodes, each corresponds to a node in $G$. In this way, the encoding tree of height 1 simply describes the fact that $G$ has $|V|$ nodes, and makes no assumptions about higher-order structures, such as communities. Figure 4 (b) shows an example of an encoding tree of height 1. On the other hand, an encoding tree of height 2 has an intermediate layer between the root node and the leaf nodes. This intermediate layer describes the 2nd-order structures, i.e., communities

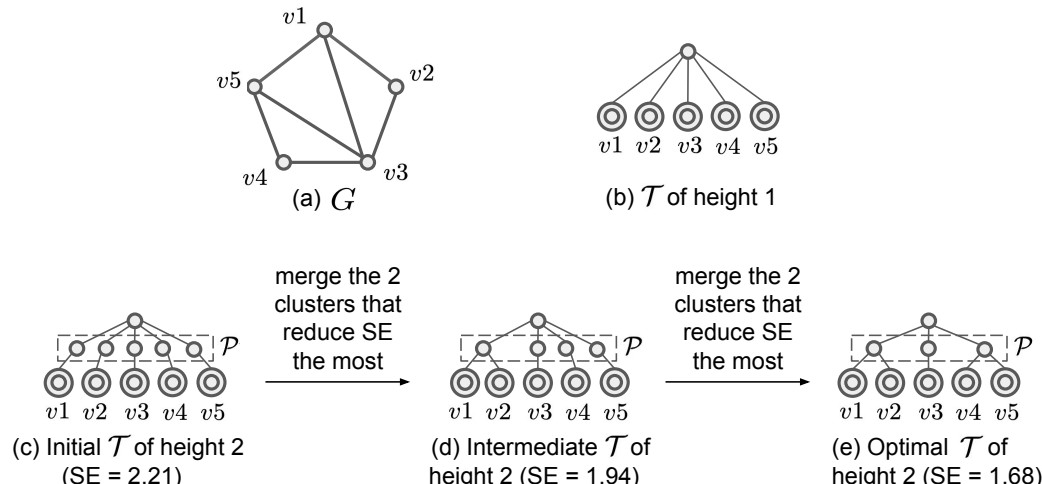

Figure 4: Examples of single-relational graph, encoding tree, and 2D SE minimization. (a) is a single-relational graph $G$. (b) is the encoding tree of height 1, which encodes the 1st-order structures, i.e., nodes, in $G$. (c) - (e) demonstrate how 2D SE minimization detects the 2nd-order structures, i.e., communities, in $G$. Initially, each node in $G$ is assigned to its own cluster. $\mathcal{P}$ in (c) shows the initial clusters. Following the vanilla greedy 2D SE minimization algorithm Li and Pan [2016], at each step, any two clusters that would reduce SE the most are merged. Eventually, the optimal encoding tree of height 2, as shown in (e), is associated with the minimum possible SE value, and encodes the communities, in $G$. $\mathcal{P}$ in (e) shows the detected communities.

in the graph. Since there are different ways to partition the nodes in $G$, a $G$ can have many encoding trees of height 2. In the task of community detection, the goal is to find the optimal encoding tree of height 2, i.e., the one that is associated with the minimized SE. Figures 4 (c) - (e) show examples of encoding trees of height 2, among which (e) is the optimal one.

Figures 4 (c) - (e) illustrate community detection through 2D SE minimization. Initially, each node in $G$ is assigned to its own cluster. $\mathcal{P}$ in (c) shows the initial clusters. Following the vanilla greedy 2D SE minimization algorithm Li and Pan [2016], at each step, any two clusters that would reduce SE the most are merged. Eventually, the optimal encoding tree of height 2, as shown in (e), is associated with the minimum possible SE value, and encodes the communities, in $G$. $\mathcal{P}$ in (e) shows the detected communities.

## C   EXAMPLES OF 2D MRSE MINIMIZATION

We provide examples of multi-relational graph, encoding trees, and the 2D MrSE minimization process in Figure 5.

As an example of multi-relational random surfing, consider $G'$ and $\mathbf{A}'$ as shown in Figure 5 (a) and (b). At each step of the multi-relational surfing on $G'$, the surfer follows $\mathbf{A}'$ to randomly and jointly decide which neighboring node to visit as well as which relation to use. E.g., assume that the

surfer is at node $v_1$. Through relation $R_1$, they can choose to visit $v_2$ or $v_5$, as $\mathbf{A}'_{2,1,R_1} = 1$ and $\mathbf{A}'_{5,1,R_1} = 1$. Similarly, through $R_2$, they can choose to visit $v_2$ or $v_5$, as $\mathbf{A}'_{2,1,R_2} = 1$ and $\mathbf{A}'_{5,1,R_2} = 1$. Finally, through $R_3$, the surfer can choose to visit $v_3$, as $\mathbf{A}'_{3,1,R_3} = 1$. In this manner, the surfer takes an infinite long random walk on $G'$.

The 2D MrSE minimization process is similar to the 2D SE minimization, shown in Figure 4. The distinction is that the proposed MrSE, instead of SE, is utilized to determine which two clusters to merge at each step.

## D   HIERARCHICAL 2D MRSE MINIMIZATION

Inspired by how Cao et al. [2024] hierarchically minimizes 2D SE, we propose to speed up Algorithm 1 with hierarchical graph partitioning. Algorithm 2 shows our hierarchical 2D MrSE minimization algorithm. Instead of simultaneously considering the entire $G'$, Algorithm 2 minimizes the MrSE of one subgraph of size $n$ at a time (lines 5-13). After minimizing the MrSE values for all subgraphs, the process continues by treating the clusters formed in the last iteration as nodes to be merged in the subsequent iteration (lines 3-4). Such a process terminates after all nodes are considered simultaneously (lines 14-15). If, at some point, it becomes impossible to merge nodes within any subgraph,

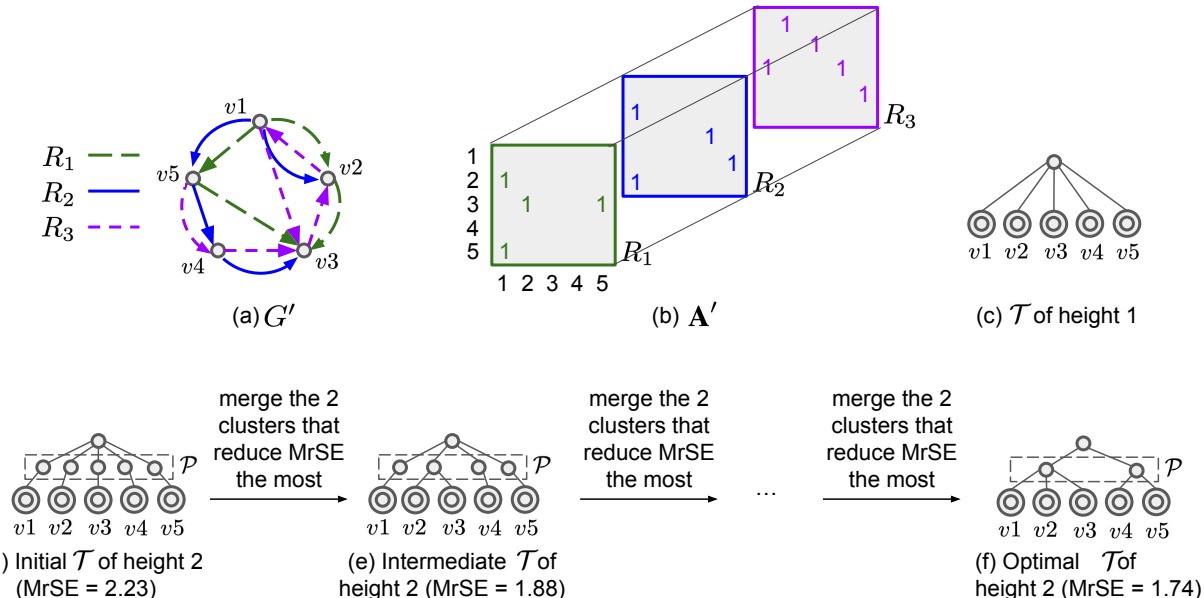

**Figure 5:** Examples of multi-relational graph, encoding tree, and 2D MrSE minimization. (a) is a multi-relational graph $G'$. (b) is the adjacency tensor of $G'$. (c) is the encoding tree of height 1, which encodes the 1st-order structures, i.e., nodes, in $G'$. (d) - (f) demonstrate how 2D MrSE minimization detects the 2nd-order structures, i.e., communities, in $G'$. Initially, each node in $G'$ is assigned to its own cluster. $\mathcal{P}$ in (d) shows the initial clusters. Following our proposed 2D MrSE minimization algorithm (Algorithm 1), at each step, any two clusters that would reduce MrSE the most are merged. Eventually, the optimal encoding tree of height 2, as shown in (f), is associated with the minimum possible MrSE value, and encodes the communities, in $G'$. $\mathcal{P}$ in (f) shows the detected communities.

we augment the parameter $n$ to encompass a greater number of nodes within the same subgraph (lines 16-17). This adjustment allows for the possibility of merging additional nodes.

The overall running time complexity of Algorithm 2 is reduced from $O(|V||E'|)$ to $O(|V_g||E_g'|) < O(n^3)$, where $|V_g| = n$ is the size of one subgraph and $|E_g'| < n^2$ is the number of edges in one subgraph.

## E   DERIVATION OF EQUATION (7)

Based on Equation (5), merging $\alpha_{o_1}$ and $\alpha_{o_2}$ into $\alpha_n$ only affects the MrSE values associated with $\alpha_{o_1}, \alpha_{o_2}, \alpha_n$, and their children. We denote the children of $\alpha_{o_1}, \alpha_{o_2}$, and $\alpha_n$ as $\Gamma_1 = \{\gamma | \gamma \in \mathcal{T}, \gamma^- = \alpha_{o_1}\}$, $\Gamma_2 = \{\gamma | \gamma \in \mathcal{T}, \gamma^- = \alpha_{o_2}\}$, and $\Gamma_3 = \{\gamma | \gamma \in \mathcal{T}, \gamma^- = \alpha_n\} = \Gamma_1 \cup \Gamma_2$, respectively.

We have

$$\Delta \text{MrSE}_{\alpha_{o_1}, \alpha_{o_2}} = \text{MrSE}_{new} - \text{MrSE}_{old}$$

$$= \underbrace{-p'_{\to \alpha_n} \log p'_{\alpha_n} - \sum_{\gamma \in \Gamma_3} \mathbf{x}'_{T_\gamma} \log \frac{\mathbf{x}'_{T_\gamma}}{p'_{\alpha_n}}}_{\textcircled{1}}$$

$$+ \underbrace{p'_{\to \alpha_{o_1}} \log p'_{\alpha_{o_1}} + \sum_{\gamma \in \Gamma_1} \mathbf{x}'_{T_\gamma} \log \frac{\mathbf{x}'_{T_\gamma}}{p'_{\alpha_{o_1}}}}_{\textcircled{2}} \quad (8)$$

$$+ \underbrace{p'_{\to \alpha_{o_2}} \log p'_{\alpha_{o_2}} + \sum_{\gamma \in \Gamma_2} \mathbf{x}'_{T_\gamma} \log \frac{\mathbf{x}'_{T_\gamma}}{p'_{\alpha_{o_2}}}}_{\textcircled{3}} .$$

Further, we have

$$\textcircled{1} + \textcircled{2} + \textcircled{3} = -p'_{\alpha_{o_1}} \log \frac{p'_{\alpha_{o_1}}}{p'_{\alpha_n}} - p'_{\alpha_{o_2}} \log \frac{p'_{\alpha_{o_2}}}{p'_{\alpha_n}}. \quad (9)$$

Plugging Equation (9) into Equation (8) concludes the derivation of Equation (7).

**Algorithm 2:** Hierarchical 2D MrSE minimization.

---

**Input:** Multi-relational graph $G' = (V, E', R)$,
       sub-graph size $n$
**Output:** An optimal encoding tree $\mathcal{T}$ of height 2

1   Initialize $\mathcal{T}$, s.t. for each node $v \in V$, add two nodes,
   i.e., $\alpha$ and $\alpha^-$, to $\mathcal{T}$. $\alpha$ is a leaf node of $\mathcal{T}$ and
   $T_\alpha = \{v\}$. $\alpha^-$ is the parent of $\alpha$ and $h(\alpha^-) = 1$

2   **while** *True* **do**

3      $\mathcal{P} \leftarrow (\alpha | \alpha \in \mathcal{T}, h(\alpha) = 1)$

4      $\{\mathcal{P}_g\} \leftarrow$ consecutively remove the first
      $min(n, \text{size of the remaining part of } \mathcal{P})$ clusters
      from $\mathcal{P}$ that form a set $\mathcal{P}_g$

5      **for** $\mathcal{P}_g \in \{\mathcal{P}_g\}$ **do**

6        // minimize the MrSE of one subgraph

7        $V_g \leftarrow$ all graph nodes $v \in V$ that are associated
        with the clusters in $\mathcal{P}_g$

8        $E'_g \leftarrow \{e \in E', \text{both endpoints of } e \in V_g\}$

9        $G'_g \leftarrow (V_g, E'_g, R)$

10       $\mathcal{T}_g \leftarrow$ construct a new encoding tree that
        contains $\mathcal{P}_g$ and the leaf tree nodes of $\mathcal{T}$ that
        are associated with $\mathcal{P}_g$

11       $\mathcal{T}'_g \leftarrow$ run 2D MrSE minimization (Algorithm
        1) on $G'_g$, with the initial encoding tree set to
        $\mathcal{T}_g$

12       $\mathcal{P}'_g \leftarrow (\alpha | \alpha \in \mathcal{T}'_g, h(\alpha) = 1)$

13       Update $\mathcal{T}$ with $\mathcal{P}'_g$

14      **if** $|\{\mathcal{P}_g\}| = 1$ **then**

15        Break

16      **if** $\mathcal{P}$ *is the same as at the end of last iteration* **then**

17        $n \leftarrow 2n$

18   **return** $\mathcal{T}$

---

## F   NODE CLUSTERING EXPERIMENTAL SETTING

We adopt a consistent embedding dimension of 64 for all embedding-based methods. For SpectralNet, we adopt a three-layer architecture of [512, 256, 64]. For node2vec and metapath2vec, we set the walk length to 100, context size to 7, walks per node to 5, number of negative samples to 5, and number of workers to 6. For all methods based on deep learning, we configure the learning rate to be 0.001 and the number of training epochs to be 200, incorporating an early stopping mechanism with patience of 50 epochs. Given the high density of the ACM and DBLP datasets, we adopt hierarchical 2D minimization (Algorithm 2) for MrSE. This approach is faster compared to the standard 2D minimization (Algorithm 1) when applied to dense graphs. Similarly, for SE and RSSE, we apply the hierarchical 2D minimization proposed by Cao et al. [2024] instead of the vanilla 2D minimization in Li and Pan [2016] for the ACM

Table 5: Node clustering ACC (%). '/' indicates that SpectralNet fails to run on ACM.

| Method | Input | ACM | DBLP | IMDB |
|---|---|---|---|---|
| k-means | $X$ | 31.97 | 28.52 | 28.94 |
| DMoN | $G, X$ | 35.07 | 6.48 | 3.62 |
| DGI | $G, X$ | 35.37 | 50.70 | **48.50** |
| SpectralNet | $G$ | / | 11.98 | 19.10 |
| node2vec | $G$ | 35.21 | 48.73 | 42.24 |
| SE | $G$ | 44.29 | 68.66 | 6.57 |
| RSSE (ours) | $G$ | 44.33 | 68.05 | 6.59 |
| DMoN | $G', X$ | 16.73 | 17.18 | 9.16 |
| DGI | $G', X$ | 71.57 | 54.47 | 35.25 |
| DMGI | $G', X$ | 55.64 | 31.23 | 37.82 |
| SpectralNet | $G'$ | / | 45.45 | 5.80 |
| metapath2vec | $G'$ | 69.72 | 66.84 | 44.09 |
| MrSE (ours) | $G'$ | **77.72** | **72.70** | 6.81 |

and DBLP datasets. We set the sub-graph size $n$ to 800 and 100 for the ACM and DBLP datasets, respectively.

## G   NODE CLUSTERING ACC

Table 5 presents the node clustering ACC. Our proposed MrSE scores the highest on the ACM and DBLP datasets, outperforming strong baselines including those that leverage node features in addition to graph structure. The MrSE shows suboptimal performance when applied to the extremely sparse IMDB dataset, likely due to the loss of structural information resulting from the stochasticity adjustments. Furthermore, MrSE proves to be a more effective tool in deciphering the community structures of multi-relational graphs compared to SE. This is evident as MrSE outperforms SE on two of three datasets and performs comparably with SE on the third dataset. Meanwhile, some methods with relatively low NMI and ARI achieve high ACC. We believe that this is related to the setting of the expected number of clusters for these methods. Specifically, as discussed in Section 4.2, for representation-learning models including node2vec, metapath2vec, DGI, and DMGI, we perform k-means clustering after learning representations, setting the number of clusters to the ground truth to obtain final community predictions. We observed that the ACC scores of these methods are sensitive to changes in the number of clusters, while the NMI scores are relatively more stable. For instance, altering the number of clusters to 50 causes the ACC of DGI ($G'$, $X$) to drop from 71.57% to 13.42% and its NMI from 47.03% to 34.74%. Similarly, the ACC of metapath2vec decreases from 69.72% to 18.94%, and its NMI from 47.51% to 29.13%.

Table 6: Statistics of the social event detection datasets. M, U, UM, H, N, and S denote message, sender, user mention, hashtag, named entity, and semantic, respectively. 'combine' indicates the single-relational edges reduced from the multi-relational ones. The statistics are presented in terms of averages. Detailed data splits can be found in Cao et al. [2024].

| Dataset | $|G'|$ | $|V|$ | $R$ | Sparsity | $|Y|$ |
|---------|------|------|-----|----------|------|
| Event2012 (avg.) | 21 | 2,314 | M-U-M | >99.99 | 37 |
| | | | M-UM-M | 99.85 | |
| | | | M-H-M | 99.60 | |
| | | | M-N-M | 96.17 | |
| | | | M-S-M | 97.32 | |
| | | | combine | 93.98 | |
| Event2018 (avg.) | 16 | 3,137 | M-U-M | 99.88 | 25 |
| | | | M-UM-M | 99.45 | |
| | | | M-H-M | 98.05 | |
| | | | M-N-M | 98.05 | |
| | | | M-S-M | 98.79 | |
| | | | combine | 94.76 | |

# H SOCIAL EVENT DETECTION DATA STATISTICS

Table 6 shows the statistics of the social event detection data. Given that all compared methods are unsupervised and exclusively utilize the test data, we limit our presentation of statistics to the test sets in Table 6.

# I MULTI-RELATIONAL MESSAGE GRAPH CONSTRUCTION

We create multi-relational message graphs by distinguishing message correlations stemming from shared senders, mentioned users, hashtags, named entities, and similar natural language semantics. To achieve this, individual single-relational graphs are constructed for each correlation type. Additionally, a combined graph is formed by consolidating various correlation types into a unified representation. In this consolidation, multiple edges of different correlation types between the same pair of nodes are reduced into a single edge. Following this, all individual single-relational graphs are concatenated along the relation axis, completing the construction of a multi-relational message graph.