# OpenReview forum: "Multi-Relational Structural Entropy"
_auai.org/UAI/2024/Conference — UAI 2024 poster_

### Official Review · Reviewer_NfKv · 2024-03-20

**Q2-1 Originality-Novelty:** 2
**Q2-2 Correctness-Technical Quality:** 3
**Q2-5 Clarity Of Writing:** 3

**Q1 Summary And Contributions:**

This paper generalizes the notion of structural entropy to multiplex networks (e.g., where node pairs can have different types of interactions). The authors propose an algorithm to minimize the MSE (but only when the encoding tree has height 2).
Experiments on synthetic and real data show that this algorithm performs well on graph analysis tasks, such as community detection.

**Q2-3 Extent To Which Claims Are Supported By Evidence:**

3: Good: the main claims are supported by convincing evidence (in the form of adequate experimental evaluation, proofs, (pseudo-)code, references, assumptions).

**Q2-4 Reproducibility:**

4: Excellent: key resources (e.g. proofs, code, data) are available and key details (e.g. proof sketches, experimental setup) are comprehensively described for competent researchers to confidently and easily reproduce the main results.

**Q3 Main Strengths:**

The extension of the structural entropy to multiplex networks appears novel and relevant to analyse complex networks. The proposed algorithm outperforms other community detection methods on benchmarks.

**Q4 Main Weakness:**

Experiments on synthetic data sets (generated from multi-layer stochastic block models for example) would have been a nice addition, to (i) show how Algorithm 1 performs with respect to the known thresholds for detectability/recovery of communities in SBMs (ii) show if Algorithm 1 performs if some layer are non-informative.

**Q5 Detailed Comments To The Authors:**

* Definition 2.1 might be hard to read, if place permits maybe a Figure could help the reader understand. I find it simpler to say that each leaf \ell of T is associated with T_{\ell} \subset V, where the T_{\ell} form a (non-overlapping) partition of V. Then for a node \alpha in the tree, T_{\alpha} is the union of the T_{\alpha'} where \alpha' runs over the children of \alpha.

* Def 2.2: log, vol should not be in italics.

* In Equation (1): \lambda is not defined, and vol(\lambda) is simply 2 |E|.
vol(\alpha) should be vol( T_{\alpha} ).
The formula for g_{\alpha} and vol(S) for S \subset V should be given (outside of the definition).

* Proposition 3.1: Equation 1 --> Equation (1) (repeated several times in the text)

* Proof of Proposition 3.1: P(i,j) and P(j \cond i) are not defined. But I think this proof could be shorter by simply writing the definition of g_{\alpha} and using vol(\lambda) = 2|E|.

* p.5 : 'set to << |V|' --> o(|V|)

* Appendix E: the results of accuracy are very different than NMI / ARI, and there is no clear pattern (for example: on ACM, the accuracy of SE is 44% but NMI / ARI are ~3%. In contrast, the accuracy of DMon is 17% while NMI is 21). Accuracy tends to be higher but not always. Is there a reason for that?
** Are all the graphs considered strongly connected?
** What is the predicted number of communities of each algorithm on the 3 data sets? (some metrics are more sensitive to over-fit or under-fit, see e.g., [1]).

[1] Gösgens, M. M., Tikhonov, A., & Prokhorenkova, L. (2021). Systematic analysis of cluster similarity indices: How to validate validation measures. In International Conference on Machine Learning (pp. 3799-3808). PMLR.

**Q9 Complying With Reviewing Instructions:**

Yes

---

> ### Author Rebuttal · Authors · 2024-04-08
>
> ***Response to Reviewer NfKv***
>
> We thank the reviewer for their time and effort in reviewing our paper and providing valuable comments. We address the reviewer comments below:
>
> > Adding figures to help the readers understand Definition 2.1
>
> We include supplementary examples of Definition 2.1 (encoding trees), in Figures 4 and 5 (https://anonymous.4open.science/api/repo/MSE-800F/file/examples.pdf). Additionally, these figures provide examples of the other core concepts of our work, including multi-relational graphs, 2D SE minimization, and 2D MSE minimization.
>
> > Why do some methods with relatively low NMI / ARI achieve high Acc?
>
> We believe that this is related to the setting of the expected number of clusters for the representation-based methods. Specifically, for representation-learning models including node2vec, metapath2vec, DGI, and DMGI, we perform k-means clustering after learning representations, setting the number of clusters to the ground truth (3, 4, 3 for ACM, DBLP, and IMDB, respectively) to obtain final community predictions. We observed that the ACCs of these methods are sensitive to changes in the number of clusters, while NMIs remain relatively stable. For instance, altering the number of clusters to 50 causes the ACC of DGI ($G’$, $X$) to drop from 71.57% to 13.42% and its NMI from 47.03% to 34.74%. Similarly, the ACC of metapath2vec decreases from 69.72% to 18.94%, and its NMI from 47.51% to 29.13%. We present more results in https://anonymous.4open.science/api/repo/MSE-800F/file/node_clustering_DGI_metapath2vec.pdf. On the other hand, for SpectralNet and DMoN, the number of clusters is a hyperparameter and is tuned to 64, which yields the best NMI. The number of clusters predicted by SE, RSSE, and MSE for ACM, DBLP, and IMDB are as follows: SE predicts 3, 23, and 612 clusters, respectively; RSSE predicts 3, 27, and 529 clusters, respectively; MSE predicts 32, 64, and 927 clusters, respectively. Notably, SE, RSSE, and MSE do not require a predefined number of clusters, offering an advantage over representation-based methods. Additionally, SE, RSSE, and MSE tend to predict fewer clusters for dense graphs like ACM and more clusters for sparse graphs like IMDB. Moreover, MSE identifies a larger number of subtle communities compared to SE and RSSE.
>
> The ACM and IMDB datasets are strongly connected, while the DBLP dataset (specifically, its A-P-A relation) is not. However, after examining the experimental results, we consider the number of clusters, rather than connectivity, as the primary factor contributing to the high ACC of representation-based methods. We appreciate the reviewer for bringing this up and will include these analysis in the draft.
>
> > Additional experiments on synthetic multi-layer stochastic block models (SBMs)
>
> We acknowledge the reviewer's suggestion regarding the evaluation of our method's performance in recovering artificially constructed communities. We appreciate the reviewer’s constructive feedback and will consider adding such experiments in future revisions.
>
> > Presentation issues
>
> We will supplement the draft with the definitions of $vol(\cdot)$, $P(i, j)$, and $P(j|i)$. In addition, we will fix the rest of the presentation issues as pointed out by the reviewer.

---

### Official Review · Reviewer_VVXP · 2024-03-22

**Q2-1 Originality-Novelty:** 2
**Q2-2 Correctness-Technical Quality:** 3
**Q2-5 Clarity Of Writing:** 3

**Q1 Summary And Contributions:**

This study introduces the heterogeneity in the graph relations into structural entropy (SE). To this end, this manuscript first defines a random surfing-based equivalent definition of the SE, and subsequently this manuscript introduces the heterogeneity to the SE. They name the proposed method multi-relational structural entropy (MSE). Also they consider a greedy algorithm to extract an essential structure from a multi-relational graph by minimizing MSE.

**Q2-3 Extent To Which Claims Are Supported By Evidence:**

2: Fair: the main claims are somewhat supported by evidence (but the experimental evaluation may be weak, or does not match entirely with the claims, important baselines may be missing, proofs contain important ideas but lack rigor, algorithmic details are only discussed superficially, references are imprecise, assumptions are not sufficiently motivated or explicated, etc.).

**Q2-4 Reproducibility:**

4: Excellent: key resources (e.g. proofs, code, data) are available and key details (e.g. proof sketches, experimental setup) are comprehensively described for competent researchers to confidently and easily reproduce the main results.

**Q3 Main Strengths:**

Equivalent formulation of SE based on random-surfing is interesting.
More rich information over a graph can be used by the proposed approach.

**Q4 Main Weakness:**

Lack of theoretical insights. No clear example.

**Q5 Detailed Comments To The Authors:**

- It is difficult to understand what the SE/MSE is doing to measure the discrepancy between two graphs. Particularly, I couldn't grasp the authors' point in Figure 1. Even though (b) and (d) show the essential structures obtained, why do the authors claim that MSE is better than SE?
- MSE usually means mean squared error in machine learning. It might be better to use a different abbreviation like "MrSE."
- We can easily compute the structural entropy for each layer of the multilayer graph and then average them; is MSE different from the SE averaged over different types of relations? If so, in what sense the proposed MrSE is superior?
- Classical Shannon entropy in information theory has many useful properties. Does MSE have similar properties? Is there any theory behind it?
- Extracting essential information from the graph using entropy is interesting. While this study evaluates the results using sparsity and NMI, is there a practical example demonstrating the extracted information? I'm interested in a simple, real-world example of the extraction.

**Q9 Complying With Reviewing Instructions:**

Yes

---

> ### Author Rebuttal · Authors · 2024-04-08
>
> ***Response to Reviewer VVXP***
>
> We thank the reviewer for their time and effort in reviewing our paper and providing valuable comments. We agree that the proposed definitions and algorithms could benefit from more examples. To address this, we have incorporated illustrative figures and runnable examples for better elucidation. In addition, we would like to highlight that MSE provides a novel and effective tool for interpreting multi-relational graphs, and has many potential applications. We hope these will improve the reviewers' impression and rating of our work.
>
> > What is the SE / MSE doing? How does Figure 1 show that MSE is superior to SE?
>
> SE and MSE are metrics that measure the amount of information contained in a single-relational graph $G$ and a multi-relational $G’$, respectively. Minimizing 2D SE and 2D MSE reveal the community structures in $G$ and $G’$, respectively.
>
> MSE is superior as it explores the relation-wise structures in addition to the node-wise structures. Figure 1 aims to show that to decode the community structures from a multi-relational $G’$, MSE can be directly applied, while SE requires $G’$ to be preprocessed into a single-relational graph $G$, which leads to information loss. As a result, SE and MSE detect different communities. For example, SE detected communities $(v_1, v_2)$, $(v_3)$, and $(v_4, v_5)$, as shown in $\mathcal{P}$ in Figure 1 (b), while MSE detected communities  $(v_1, v_2, v_3)$, and $(v_4, v_5)$, as shown in $\mathcal{P}$ in Figure 1 (d). Admittedly, without the ground truth communities, one can not tell from Figure 1 if the communities detected by the MSE are more accurate. To show that MSE is indeed superior to SE, we conducted experiments in Section 4 and observed that MSE offers a more insightful interpretation of the structures of multi-relational graphs and enhances the performance on various real-world multi-relational graphs.
>
>
> > Real-world examples
>
> We illustrate community detection through 2D SE minimization and 2D MSE minimization in Figures 4 and 5 (https://anonymous.4open.science/api/repo/MSE-800F/file/examples.pdf), respectively. For 2D MSE minimization, initially, each node in $G'$ is assigned to its own cluster. $\mathcal{P}$ in Figure 5 (d) shows the initial clusters. Next, following our proposed 2D MSE minimization algorithm, at each step, any two clusters that would reduce MSE the most are merged. Eventually, the optimal encoding tree of height 2, as shown in Figure 5 (f), is associated with the minimum possible MSE value, and encodes the communities in $G'$. $\mathcal{P}$ in Figure 5 (f) shows the detected communities. The 2D SE minimization, shown in Figures 4 (c) - (e), is similar, except that the process is guided by SE rather than MSE.
>
> These examples are accompanied by runnable code (i.e., test_MSE() and test_SE_2() in https://anonymous.4open.science/r/MSE-800F/compare_SE_MSE.py) that prints the SE / MSE values and communities at each step.
>
> > Why can't MSE be replaced by averaging the SE over different types of relations?
>
> When considered separately from each other, each relation corresponds to a distinct graph with its own community structure. Although these single-relational graphs share a common number of nodes, their community structures are not aligned, which would prevent us from getting a unified community detection result. Take the $G’$ in Figure 5 (a) as a concrete example, from the single-relational graph that corresponds to relation $R_1$, we can detect communities $(v_1, v_2)$, $(v_3)$, $(v_4)$, and $(v_5)$. From the single-relational graph that corresponds to relation $R_2$, we can detect communities $(v_1)$, $(v_2)$, $(v_3)$, and $(v_4, v_5)$. From the single-relational graph that corresponds to relation $R_3$, we can detect two communities $(v_1\, v_2\, v_3)$, and $(v_4, v_5)$. It is unclear how to address the disagreements. The proposed MSE, in contrast, detects communities that are agreed by all the relations.
>
> > ‘MrSE’ as an alternative name
>
> We will replace ‘MSE’ with ‘MrSE’.
>
> > Preferable properties of SE and MrSE
>
> Similar to Shannon's Entropy, SE shares the property of additivity. Additionally, SE has properties such as *network dependency* (fails to hold for the other measures of complexity of networks), i.e., $\mathcal{H}^{(k)}(G)$ is a function of the sizes of $G$; *locality*, i.e., by minimizing SE, none of the communities can expand to encompass the entire network; *local and incremental computability*, i.e., rather than recalculating SE from scratch each time there is a modification to the network, one can update the computation based on the specific changes that occur. Please refer to Section VIII of [1] for proofs. As the multi-relational extension of SE, MrSE also has the above properties.
>
> [1] Li, A., et al., "Structural information and dynamical complexity of networks." 2016 IEEE Transactions on Information Theory.

---

### Official Review · Reviewer_bNhs · 2024-03-22

**Q2-1 Originality-Novelty:** 3
**Q2-2 Correctness-Technical Quality:** 3
**Q2-5 Clarity Of Writing:** 3

**Q1 Summary And Contributions:**

The paper develops a multi-relational version of structural entropy.
Structural entropy is a measure for structural information of graphs, and one of many approaches to define a measure based on the Shannon entropy.
The authors generalize this approach to multi-relational graphs, and utilize random surfing and a greedy algorithm for graph learning.
Experiments and real-world examples illustrate the results.

**Q2-3 Extent To Which Claims Are Supported By Evidence:**

3: Good: the main claims are supported by convincing evidence (in the form of adequate experimental evaluation, proofs, (pseudo-)code, references, assumptions).

**Q2-4 Reproducibility:**

3: Good: key resources (e.g. proofs, code, data) are available and key details (e.g. proofs, experimental setup) are sufficiently well-described for competent researchers to confidently reproduce the main results.

**Q3 Main Strengths:**

The paper studies an interesting problem with a creative approach. The presentation is mostly clear and rigorous.

**Q4 Main Weakness:**

Some background needs expansion for a wider audience, for example the multi-relational graph, the encoding tree or random surfing could benefit from examples.
Some assumptions are not discussed, for example whether a graph allows a representation by an encoding tree or the parameters of primitivity adjustment.

**Q5 Detailed Comments To The Authors:**

1) Please discuss the assumption whether a graph allows a representation by an encoding tree.
2) Are there other options for creating the transition probability matrix from an adjacency than via column normalization?
3) In p.3 you write "Equation 1 can be rewritten as:...", but I understand that you prove this in Proposition 3.1. I would suggest to clarify the presentation here (if I am not misunderstanding something here).
4) Discuss the choices of E and c in primitivity assessment, or give further references.
5) I would consider $A'$ to be an adjacency tensor in p.4, as it is clearly not a matrix?
6) p.5: arabasi -> Barabasi

**Q9 Complying With Reviewing Instructions:**

Yes

---

> ### Author Rebuttal · Authors · 2024-04-08
>
> ***Response to Reviewer bNhs***
>
> We thank the reviewer for their time and effort in reviewing our paper and providing valuable comments. We address the reviewer comments below:
>
> > Adding examples of the encoding tree, multi-relational graph, and random surfing
>
> We provide examples of the multi-relational graph, encoding tree, as well as the 2D SE / MSE minimization in Figures 4 and 5 (https://anonymous.4open.science/api/repo/MSE-800F/file/examples.pdf).
> For an example of multi-relational random surfing, please consider $G’$ and $\textbf{A}'$ as shown in Figures 5 (a) and (b). At each step of the multi-relational surfing on $G’$, the surfer follows $\textbf{A}'$ to randomly and jointly decide which neighboring node to visit as well as which relation to use. E.g., assume that the surfer is at node $v_1$. Through relation $R_1$, they can choose to visit $v_2$ or $v_5$, as $A_{2,1,R_1} = 1$ and $A_{5,1,R_1} = 1$. Similarly, through $R_2$, they can choose to visit $v_2$ or $v_5$, as $A_{2,1,R_2} = 1$ and $A_{5,1,R_2} = 1$. Finally, through $R_3$, the surfer can choose to visit $v_3$, as $A_{3,1,R_3} = 1$. In this manner, the surfer takes an infinite long random walk on $G’$.
> We will add the above examples to the draft.
>
> > Whether a graph allows a representation by an encoding tree
>
> A graph can always be represented by encoding trees. This is because an encoding tree is essentially a description of a graph’s structure. For a graph $G$, the encoding tree of height 1 is unique, containing one root node and $|V|$ leaf nodes, each corresponds to a node in $G$. In this way, the encoding tree of height 1 simply describes the fact that $G$ has $|V|$ nodes, and makes no assumptions about higher-order structures, such as communities. Figure 4 (b) (https://anonymous.4open.science/api/repo/MSE-800F/file/examples.pdf) shows an example of an encoding tree of height 1. On the other hand, an encoding tree of height 2 has an intermediate layer between the root node and the leaf nodes. This intermediate layer describes the 2nd-order structures, i.e., communities in the graph. Since there are different ways to partition the nodes in $G$, a $G$ can have many encoding trees of height 2. In the task of community detection, the goal is to find the optimal encoding tree of height 2, i.e., the one that is associated with the minimized SE. Figures 4 (c) - (e) show examples of encoding trees of height 2, among which (e) is the optimal one.
>
> > Are there other options for creating the transition matrix from an adjacency than via column normalization?
>
> Yes. One example is the popular HITS algorithm [1] and its many variants. HITS simultaneously considers each node as a hub and an authority, it decomposes the transition process into two mutually reinforcing operations, i.e., authorities pointed by hubs and hubs point to authorities. Rather than normalizing the adjacency matrix $A$, it models the ‘authorities pointed by hubs’ operation with $A$ and the ‘hubs point to authorities’ operation with $A^\top$. Consequently, $AA^\top$ is the transition matrix for propagating the authority scores, $\textbf{x}$, while $A^{\top}A$ is the transition matrix for propagating the hub scores, $\textbf{y}$. [2] shows that HITS and the weight normalization-based methods such as PageRank are closely correlated.
>
> [1] Kleinberg, J. M. "Authoritative sources in a hyperlinked environment." In JACM 1999.
>
> [2] Ding, C., et al. "PageRank, HITS and a unified framework for link analysis." In SIGIR 2002.
>
> > Parameters of primitivity adjustment
>
> Our proposed RSSE adopts $E = 1/|V|\textbf{e}\textbf{e}^\top$ and c = 0.85 following the PageRank algorithm (as discussed in Section 3.1, ideally, c should balance two needs, i.e., small enough to make the Power Method converge fast, and large enough to keep the intrinsic structures of $G$. We propose to explore the best strategy for choosing c in the future).
> On the other hand, our MSE adopts $\textbf{E}' = 1/|V|\textbf{1}$, where $\textbf{1}$ is a $|V|\times|V|\times|R|$ all-ones matrix, and c = 0.85. These choices follow the same intuition as RSSE. Specifically, the $\textbf{E}'$ value specifies that for any relation $r \in R$, the surfer has equal chances to teleport to any of the objects. We will add the above discussion to the draft.
>
> > Presentation issues and typos
>
> As pointed out by the reviewer, $\textbf{A}'$ is a tensor rather than a matrix. We will fix this and the other presentation issues per the reviewer’s comments.

---

### Official Review · Reviewer_VAgV · 2024-03-28

**Q2-1 Originality-Novelty:** 3
**Q2-2 Correctness-Technical Quality:** 3
**Q2-5 Clarity Of Writing:** 4

**Q10 Ethical Concerns:**

None.

**Q1 Summary And Contributions:**

The paper proposes MSE, a metric of multi- relational graph structural information. Based on the original definition of SE from the viewpoint of random surfing, a definition of MSE is derived from random surfing on multi-relational graphs. Furthermore, the paper we introduces a 2D MSE minimization algorithm designed to unveil communities within complex graphs. Results from experiments on both synthetic and real-world graphs, including movie, publication, and social message networks, demonstrate that the proposed MSE is a useful metric for assessing and unraveling structural information within multi-relational graphs. MSE is shown to exhibit strong performance in two tasks, namely multi-relational node clustering and social event detection.

**Q2-3 Extent To Which Claims Are Supported By Evidence:**

3: Good: the main claims are supported by convincing evidence (in the form of adequate experimental evaluation, proofs, (pseudo-)code, references, assumptions).

**Q2-4 Reproducibility:**

3: Good: key resources (e.g. proofs, code, data) are available and key details (e.g. proofs, experimental setup) are sufficiently well-described for competent researchers to confidently reproduce the main results.

**Q3 Main Strengths:**

Structure properties of graphs are relevant for many application tasks, and the paper contributes to realistic multi-relational graphs.

The presentation is rigorous, although hard to perceive due to mathematical conciseness. A few examples would have helped, but length restriction had certainly been the obstacle.

Relevant experimental results are indeed provided, and the code is available.

**Q4 Main Weakness:**

The figures are way too small. Examples for definitions are not provided, but are actually required to help the reader understand the main points.

The relation to uncertainty comes with steady state distributions, the relevance of the work for AI systems could be better explained, however.

**Q5 Detailed Comments To The Authors:**

Nice.

**Q9 Complying With Reviewing Instructions:**

Yes

---

> ### Author Rebuttal · Authors · 2024-04-08
>
> ***Response to Reviewer VAgV***
>
> We thank the reviewer for their time and effort in reviewing our paper and providing valuable comments. We are pleased to see that our paper's contribution to multi-relational graphs and the rigor of its presentation were noted. We address the reviewer comments below:
>
> > Adding examples of the definitions
>
> To facilitate comprehension of the definitions and the core concept of our work, we include supplementary examples of encoding trees, multi-relational graphs, 2D SE minimization, and 2D MSE minimization in Figures 4 and 5 (https://anonymous.4open.science/api/repo/MSE-800F/file/examples.pdf).
>
> > Figures are too small
>
> We will resize all figures to enhance readability.
>
> > The relevance of the work for AI systems
>
> Our work extracts the essential structure of multi-relational graphs, and has many potential applications in AI systems. For instance, it can serve as a preprocessing tool, aiding in denoising input graphs. Additionally, when applied to sequential models, it enables the transformation of the higher-order graphs into sequences while preserving the hierarchical structures of the original graphs.

---

### Meta-Review · Area_Chair_zQxh · 2024-04-21

This paper generalizes the notion of structural entropy to multiplex networks (e.g., where node pairs can have different types of interactions). The authors propose an algorithm to minimize the MSE (but only when the encoding tree has height 2). Experiments on synthetic and real data show that this algorithm performs well on graph analysis tasks, such as community detection.

Note. Reviewer VVXP notes that he changed his rating to weak accept as a consequence of the authors' explanations and discussion, but I don't see the rating changed in the system. Probably an oversight.